# Sustainable Health Care Provision Worldwide: Is There a Necessary Trade-Off between Cost and Quality?

**Chhabi Lal Ranabhat** [1,2,*] and **Mihajlo Jakovljevic** [3,4,5]

1   Department of Health Administration and Promotion, College of Public Health, Eastern Kentucky University, Richmond, KY 40475, USA
2   Global Center for Research and Development (GCRD), Kathmandu 446088, Nepal
3   Institute of Advanced Manufacturing Technologies, Peter the Great St. Petersburg Polytechnic University, 195251 St. Petersburg, Russia
4   Institute of Comparative Economic Studies, Hosei University, Tokyo 194-0298, Japan
5   Department of Global Health Economics and Policy, University of Kragujevac, 34000 Kragujevac, Serbia
*   Correspondence: chhabir@gmail.com

**Abstract:** Quality health care is an essential human right, on the agenda of sustainable development and presents a challenge in the twenty-first century. There are different perspectives regarding the price and quality of health care, and it is necessary to review the quality health care issue and how it influenced by price. The aim of this study is to explore the different dimensions of health care quality, examine the association with technology, health care market characteristics, additional and optional services of health care, sustainability, and some exceptional situations. We performed the narrative review searching by key words by main search engine Google and followed by their mother publication and or any first web database. We found that health care is a service industry, needs basic standards and specialized human resources to perform the procedure, and quality health care is not associated with an extra price. The quality of health care assures sustainability. Likewise, there are some additional choices during certain procedures, and those may have different price options and would be linked with quality. So, those optional health care and basic health need to define separately.

**Keywords:** health care; quality; sustainability; price; health care market; additional health care

## 1. Introduction

Quality health care is a right for every person around the globe and providing quality health care is the responsibility of a state to its citizens [1]. There are different perspectives regarding quality health care. Quality health care increases the likelihood of desired health outcomes for individuals and populations, and should be consistent with current professional knowledge [2]. According to the Agency for Health Care Research and Quality (AHCRQ), the basic domains of quality health care are effectiveness, efficiency, equity, patient-centeredness, safety, and timeliness [3]. Similarly, quality health care ensures sustainability, which is a priority for development [4]. The World Health Organization (WHO), gives the basic principles for quality health care as the right care, at the right time, responding to the service users' needs and preferences, while minimizing harm and resource waste [5]. There are many challenges and barriers to receiving quality health care, but it is the right of every person [6]. There is considerable debate about whether quality health care comes at an additional cost. In other words, people who pay more will get quality health care, and those who are not able to extra pay will not [7].

There is much research regarding the poor quality of health care. It is the major challenge in developed and developing countries alike. In high-income countries, >10% patients were adversely affected during treatment [8], 7% of patients in hospital acquired infections and were subjected to irrational use of antimicrobials [9]. Due to the misuse and overuse of antimicrobials in health care, antimicrobial resistance is a major public

health problem [10]. The cost associated with medication errors in Thailand has been estimated at USD 1.2 billion (42 billion Thai Bhat) annually, not counting lost wages, foregone productivity or health care costs [11,12]. Nearly 40% of health care facilities in low- and middle-income countries lack improved water and nearly 20% lack sanitation; the implications for quality of care are clearly evident [13]. A study by Yaqi Yuan, Wake Forest University, USA found that in 30 countries only 17% of people were satisfied with their health care facilities [14]. A healthy debate regarding quality health care is needed, and there is a critical situation regarding health care quality, especially for people living in remote areas with poor access to the health services.

Better quality in healthcare means a systematic approach from a health care organization that monitors, assesses, and improves the standards of quality health care. The organizational system is cyclic (yearly) and needs continued improvement each fiscal year to seek a higher level of performance. Sylvia et al. highlighted that continuous improvement in health care activities converts health care organizations from inefficient traditional concepts to efficient ones that utilize new technologies and management models to perform efficiently, and hence, generate better quality results [15]. Previous studies on quality health care have mostly focused on clinical aspects such as diagnosis, treatment, patient compliance [15], behaviors of health care providers [16,17], regular follow up [18], and clinical practice for patients by specialists [19]. A study by Dixon-Woods et al. explored the three aspects of health care quality and challenges in terms of design and planning of improvement and interventions, the organizational and institutional contexts, professions and leadership, sustainability, spread, and unintended consequences [20]. Quality health care necessities the autonomy and motivation of health care providers [21]. Much research is being carried out on digitalization, organizational management, customer satisfaction, claims and payment, and disparities in health care. The rising cost of health care is combined with the twentieth century's transnational trend of systemizing the health care sector and profit maximization of health care industries [22]. Countries such as the USA spend more on health care but there are many issues with quality and price, whereas countries such as Costa Rica, Thailand and Singapore spend an average amount but provide high quality health care [23]. Health care quality and price may be satisfactory in health promotion and preventive care, but must be found in curative and palliative care too [24]. There are very few researchers comparing quality and price of healthcare [25,26]. Normally, price increases in healthcare do not occur in a vacuum and there might be some rationale for these. On the other side, health care is a specialist service industry, and quality means the fundamentally good skill and behavior of the healthcare provider, and providing such service is part of their professional ethics. The price and quality of health care needs to be assessed in a fair way. Our review explores the fundamental factors in health care quality, determinants, and sustainability. The aim of this review is to explore the different dimensions of health care quality, examine the association with technology, characteristics of health care market, additional and optional services of health care, sustainability, and identify some exceptional situations.

## 2. Methodology

We focused on quality health service with sustainability and price in four aspects: quality health care and application of technology, an overview of the health care market, optional or menu-based health care, and its influence in price and sustainability. These are the fundamental factors that determine health care access and equity [27,28]. Major key words used to search the literature were health care, quality, sustainability, price, health care market, additional health care, health care infrastructure, quality health behaviors, enablers and barriers for quality health care, exceptional conditions and health care quality, sustainability, and quality health care. We used Google for our search engine. We selected the literature in 4 steps: (1) Each the literature by key words through Google, (2) Sort the best matches by title in the first 5 web pages, (3) Use full access literature from the selected titles and (4) Use or reject papers as appropriate to our subject. There was duplication in the

results from using multiple indexing services such as PubMed, Google Scholar, CINAHL, Scopus, etc. Regarding the quality of literature, almost all papers were taken from peer reviewed journals and all are available online. The nature of our paper concerns an important public health issue, and the best-suited method could be a narrative review [29], so, we did not count the number of published papers, type of study, special preference in publications, country, region, diseases, risk factors, medical procedure, sensitivity, specificity, or time of publication. Those factors are mandatory in clinical research and systematic reviews [30], but our research did not require hard inclusion and exclusion criteria. The issues related to health care quality and price are not limited by demography, culture, region, season, or timeline. A narrative review attempts to summarise the findings that have been written on a particular topic but does not focus on numbers and trends [31,32]. Furthermore, there are no specific including and excluding criteria for sorting the literature [33,34] and in public health-debated issues, it is necessary to first explore the evidence and then recommend a bibliometric review or systematic review if necessary [35].

Narrative review is a common approach to public health issues. The purpose of the narrative review is to explore evidence on scaling up public health interventions into population-wide policy and practice, with a focus on defining and describing the processes and frameworks that support the scaling up of public health initiatives [36]. Narrative reviews can stimulate debate on public health issues and suggest further research such as creating research questions, rethinking the existing policy gap, and ultimately contribute to the sustainable health care in terms of price, quality and access of health care to people [37]. Our narrative review was mostly confined to health policy, management, different stages of project life cycles, and daily public health issues to provides a solid foundation for the development of new theoretical perspectives in this area other than the selection and coding process generally used in research [34]. The scope of our narrative review mostly critiques and summarizes a body of literature and its conclusions [38]. The major three steps were literature gathering through a key word search, appropriately reviewing the resulting papers and producing this analysis [39]. We have used this process to explore the quality and price of health care in a critical way.

## 3. Results

### 3.1. Dimensions of Health Care Quality

Health care is a service but has become a business following economic liberalization. In comparison with other commodities or goods, health care needs specialized human resources. Mwachofi 2011 summarized health care as a service for ill or sick people, and normally nobody can predict service needs, the outcome of care cannot be assured, most of the industry is dominated by nonprofit providers, and payments are made by third parties, mostly by government agencies [40]. Other market products are typically consumed on a daily basis; industry produces agency or vendor supplies and wholesalers or retailers sell those to the consumers. If we look at the health care industry, there are different departments such as patients, hospitals, specialists, nurses, pharmacies, urgent/emergency care, physical therapy, diagnostic facilities, primary care providers, payers, and researchers [41].

Quality health care can be viewed from different angles. One, it needs to make health care/service safe, patient-centered, timely, effective, efficient, and equitable. Allen-Duck 2017 described 'right theory' as doing the right thing, the right time, for the right patient, in the right way to achieve the best possible results [42]. Similarly, the quality strategies published by Centers for Medicare & Medicaid Services (CMS) stated that quality health care strategies provide assurance of safety by reducing harm caused in the delivery of care, strengthening person and family engagement as partners in their care, promoting effective prevention and treatment of chronic disease, and effective communication and coordination of care [43].

There is a large difference in quality health care from the patients' perspectives. A study by Sixma et al., 1998 concluded that there are two components that profoundly affect the quality of health services: performance components and behavioral components [44].

A meta-analysis by Hall and Dornan 1998 from 221 studies explored the relationship between patient satisfaction and medical care [45,46]. Another study from Hall 1998 from 41 observational studies showed that patient satisfaction was related to objective measures of information giving, technical and interpersonal competence, providers' partnership-building, and socioemotional behavior, such as a provider's nonverbal behavior, social conversation, and positive talk [47]. Apart from the patient satisfaction, there is a umbrella concept consisting of 11 aspects that include humaneness (65% of all studies), informativeness (50%), technical skills of the health care provider (43%), bureaucratic procedures (28%), access to and availability of health care services (27%), costs of treatment and flexibility of payment mechanisms (18%), the comfort of seating, attractiveness of waiting rooms, clarity of signs and directions, quietness and neatness of health care facilities (16%), continuity of care (6%), outcome of the health care process, in terms of usefulness or effectiveness (4%), and attention to psycho-social problems (3%) and overall quality of care (45%).

Regarding the quality of health care, there are similar findings from health care providers also. The Council of Accountable Physician Practices (CAPP) 2017 concluded that the doctor/patient relationship, evidence-based medicine, care coordination, facilities, access, technology, and preventive services are major dynamics of quality services [48]. A study by Ying Liu et al., 2021 explored six dimensions of quality health care from nursing perspectives which were task-oriented activities, staff characteristics, physical environment, human-oriented activities, pre-conditions, and patient outcomes [49].

### 3.2. Quality Health Care, Application of Technology and Price

Innovation in health care technology is linked to improved health care quality. In most cases, new technology is required to reduce the cost of human resources while also providing quick results in disease diagnosis and treatment. The application of new machines in health care settings begin after the economic evaluation of health care, which includes cost benefit, cost effectiveness and cost utility analysis. Geligns 1991 and Tabis 2005 concluded that if the new technology could not minimize the cost, had a short life span, was difficult to maintain, and had a high environmental hazard, it was not acceptable [50,51]. A study by Ben-Israel et al., 2021 found that unruptured intracranial aneurysms (UIA) was a new innovation in open surgery and application of this technology is economically feasible [52]. Similarly, a study by Mohammad ZahedulAlam et al., 2022 found that mobile health (mHealth) wan not associated with price of health care in developing countries [53]. Fundamentally, the application of new technology that does not increase the price of health care is a onetime investment with the aim of reducing the cost of health care. Research by Belfiore et al., 2018 showed that Human Body Posturizer (HPB), an innovative therapeutic tool used in lower back pain and its use reduced the cost of treatment by one-third [54]. Kos (2018) noted that new technology in health could be adopted after evaluating the health problem and current use of technology, the technical characteristics of technology, its safety, clinical effectiveness, costs and economic evaluation, ethical analysis, organizational aspects, patient and social aspects and legal aspects [55]. They concluded that new technology is seldom responsible for increasing the price of health care.

Health care is a service industry, and receiving good service is part of quality care. Kos 2018 investigated the five major costs associated with providing health care services: labor, materials, overhead, fixed costs, and variable costs [55]. Elisabeth Engl et al., 2019 concluded that the friendly behavior of health care providers is the major factor in quality care from the patient's perspective [55,56]. A systematic review by Ahmet Nacioglu 2016 showed that clear communication about patient safety enhances the quality of care, which is related to effective communication and practice of professionalism [57]. Another important factor in quality health care is physical infrastructure. A cross-sectional study of 4300 facilities from 8 countries by Leslie et al., 2018 showed that ensuring adequate space for care was available in every department and waiting room, hygiene and sanitation, greenery, and space management provided quality care and a pathway for sustainable health care [58]. An editorial by Luxon summarized the characteristics of quality health care in eight areas;

built environment or adequate infrastructure, medical equipment, access with service, technology, governance and organizational structure, staff structure and team work were the foundation of sustainable health care [59].

### 3.3. The Health Care Market, Issue of Quality Health Care, and Price

The revenue from the health care industry worldwide is projected to be about 50 billion, annual growth is 12%, with a projected market volume of USD 15,830.00 m, with most of the revenue generated in China by the end of 2022 [60]. Ignoring the few exceptions, the health care industry is similar to other industries, and most of the economic rules match [61]. As in other industries, the inflation rate, demand and supply, market competitions, and price elasticity all apply. However, the health care profession or industry is more specialized than other daily goods consumption because it requires highly skilled human resources, sophisticated diagnostic laboratories, and scientifically-proven drugs to treat disease. Martin Gaynor 2023 concluded that there are few agencies entering the market due to its complexity. A special agency responsible for monitoring and oversight of health-care markets is necessary, but the market rules are similar in health care industry as other industries [62]. Moreover, patients need special care from highly-skilled nurses and medical assistants for a speedy recovery. Branning 2016 stated that in spite of the special characteristics of the health care industry in the USA, the costs of care could be split as follows: up to 30% in physician costs, 10% in pharmaceutical cost, 21% in administrative cost and up to 36% in new innovation and technology [63]. It seems that the administrative cost is similar with other industries, while cost for new innovation and technology obviously pays back and does not necessarily increase the price, but a review by Ogura et al., 2014 concluded that to enhance the quality of care, human resource costs are higher than in other industries but somehow, these can be balanced by appropriate supply chain mechanisms [64]. Barber et al., 2021 carried out country case studies concerning health care services' price setting and regulation approaches and concluded that collective negotiations and unilateral price setting have the potential to control price levels and avoid price discrimination [65]. As a result, the main issue is not price increases, but cost adjustment in the health care industry.

### 3.4. Menu-Based Health Care and Quality

There is a practice of menu-based health service, and any client/patient can choose additional services. During a hospital stay, there might be sophisticated services for those who can afford them. These menus are mostly in private health care facilities. Some patients may want large rooms with television, Wi-Fi, a spa or sauna, physical therapy, different ways of entertainment, etc., during their hospital stay [66]. Moreover, there are several health centers and clubs that are a part of health care. Every health service provided by health facilities is not essential. They may not significantly cure diseases and they are only supportive, but Currow 2016 revealed that the additional cost/price may affect with patient outcomes, and it can be linked with quality of care [67]. There are specific clinical procedures, protocols, standards and guidelines in medical practice, and World Health Organization (WHO), frequently updates them. Heymann 1994 highlighted that those clinical protocols are keys to maintaining quality health care, and adopting those protocols is a professional ethic for health care providers [68]. Neff et al., 2009 noted that there is no mandate to follow the protocols and guidelines in those health care menus and ultimately those optional services increase the health disparity [69]. The price difference of those menu items obviously reflects the economic class, creates a situation of selection bias, and may result in catastrophic health expenditure [70].

### 3.5. Sustainable Health and Health Care Quality

Sustainability refers to meeting our own needs without compromising the ability of future generations to meet their needs, and quality care should lead to sustainable care. Sustainability can be conceptualized as an area of the quality of health care, extending the

responsibility of health services provided to patients from now to the future. A sustainable approach maintains the value of health care, directs for positive treatment outcomes and ultimately reduces the financial burden. Hovlid et al., 2012 concluded that quality health care is a prerequisite for sustainability [71]. A study by MacNeill et al., 2021 highlighted that failing the challenges of sustainable health care would result in poor quality of health care and large amount of money wasted in the USA [72]. An analysis by Clery et al., 2021 suggested that quality improvement in health care accelerated the sustainability in every perspective and maintained the good relationship between health care providers and patients [73]. Maeda et al. concluded that Universal Health Coverage (UHC) not only assures access, equity, and protection in health care but also ensures quality health care and a road to the sustainable health care [74]. A qualitative study by Anam Parand et al., 2012 explored that a Safer Patients Initiative is one of the best strategies to ensure the quality health care pathway to sustainability in health care [75].

*3.6. Expectational Situations*

There are exceptional situations that do not occur normally. The preceding facts demonstrate that high-quality health care does not always come at a high price. Nonetheless, Jacovljevic 2022 and Park 2022 found that there are some situations that become out of control for some period of time, such as global crises such as pandemics and wars, and some supply chain restrictions, a scarcity of expert service, and high market liquidation [76,77]. Zeus Aranda, Thierry Binde et al., 2022 showed that there were significantly reductions in maternal health service due to disturbances of supply chain in 37 low and middle income countries [78]. Efrat Shadmi, Yingyao Chen et al., 2020 examined 13 countries and found that during the COVID-19 pandemic there was profound disruptions of health care access, equity, and quality in primary care [79]. Roberton et al., 2020 revealed that there was a 27% increase in the under 5 mortality rate, and a 23% increase in maternal mortality due to the 2014 Ebola virus epidemic, and the influenza pandemic of 2003 [80]. Miljeteig et al., 2021 reported that there was a scarcity of human resources such as physicians and nurses and a need to pay high salaries, ethical violence in medical practices and insufficient personal protective equipment, and ultimately, a need to pay high even for primary care during the COVID-19 pandemic in Norway [81]. The above studies show that there can be significant compromises in health care access, quality, and price in exceptional situations such as pandemics, and health care management should be handled carefully.

Based on the above studies, indicators of quality health care are presented below (Table 1).

**Table 1.** Indicators of quality health care and assessment of additional cost.

| S.N. | Characteristics of Quality Health Care | Area of Health Care | Need of Additional Resource/Capital |
|---|---|---|---|
| 1 | Effective and friendly communication with patients | Communication | No need |
| 2 | Friendly behavior to the patients and use of standard protocols to care for patients | Attitude of health care provider | No need |
| 3 | Short/minimum length of stay in health care facilities | Hospital admission | No need and reduce the cost of health care |
| 4 | Standard health care setting, viz., sanitation, water, health care waste management, greenery | Water, Hygiene and Sanitation (WASH) | No need and appropriate management would work |
| 5 | Short and skillful and timely medical process | Technical skill of health care providers | Normally no need and application of new technology need the resource for first time |
| 6 | Painless medical procedures | Drug, equipment, and technical skill of health care provider | Normally no need and application of new technology for medical procedure need the additional resource for first time |
| 7 | Adequate space to check up and assurance of privacy for clients | Infrastructure and skill of health care provider | One time investment for infrastructure |
| 8 | Smart and friendly administration process that manage time and queue/line of health | Health care administration | No need extra money |

## 4. Discussion

Quality health service is associated with patient satisfaction, safety, behavior of health care providers, and smart administration processes. Moreover, it is a special and complex service because it requires expert services, patient-friendly care providers, accuracy in disease diagnosis, and drug and treatment protocols. Our study concluded that health care is a service, and the quality of the service is not associated with additional cost. A study by White C., Reschovsky J.D. 2014, Bond A.M discovered that higher-priced hospitals outperformed lower-priced hospitals on reputation-based quality measures, but performed worse on measures of excessive readmissions and patient safety indicators, such as postsurgical deaths and complications [82]. Studies by Gabor A. et al., 1966 [83], Leavitt HJ et al., 1954, [84] clearly indicated that from the demand side, patients, like other clients or consumers, believe that high prices are indicators of better quality but there is no difference in quality supplied by providers. A similar study by Vanichchinchai, Assadej 2020, concluded that there was no difference in quality from health care providers in terms of patient safety, privacy and follow-up treatment protocols, even if there is price variation in health care facilities [85].

A qualitative study by Barber et al. 2021 highlighted that price setting in health care might be associated with quality, but further studies are essential [65]. One of the determining factors for health care cost is payment for health care providers such as physicians, but a study by Eric T. Roberts, Ateev Mehrotra, and J. Michael McWilliams 2017 concluded that high- and average-priced physicians do not affect efficiency and quality care [86]. A study by Casalino LP 2014 and Crespin DJ 2016 suggested that the cost of health care can be reduced through organizational and administrative improvements without compromising the quality of care [87,88]. The hospice and long-term care organizations have more choices, and obviously sophisticated health services are related to higher prices [26]. Those findings are in line with our conclusion. As with other items and commodities, the better-quality items need more investment and incur additional costs, but health care is mostly related to human behavior, knowledge and skill, ways of communicating and mutual respect.

The items or commodities used in patient care have a cost, but they do not always necessitate a significant additional cost. Even in basic or urgent care there are many choices in terms of prices. For health promotion and wellness, there might be many options, and prices may differ and be associated with the quality of care. Moreover, in some exceptional situations, the health care industry or market may also need exceptional approaches. To summarise, quality of health care is the right of individuals, the responsibility of the state, and the duty of health care providers without compromise.

## 5. Conclusions

This study explored the different aspects of quality health care and concluded that a high price or additional payment does not assure quality or sustainability. It means that the price of health care is obviously influenced by market factors such as a supply chain, inflation, consumer purchasing power, and the situation of the health workforce. Further specific and advanced studies are essential. However, our study has some limitations. Firstly, it does not specify the price according to given quality standards. Secondly, it does not focus on the quantitative presentation of available literature using numbers, types, published times, sample size, etc. Obviously, a narrative review is less rigorous and more subjective. More research is needed to examine the cost and quality of specific health care services such as preventive, promotional, and curative care from both the patient and provider perspectives.

**Author Contributions:** C.L.R. conceptualized, designed, prepared, reviewed, and leaded the paper and M.J. updated the manuscript in methodology and results. Both authors contributed fulfilling ICJME conditions for full authorship. All authors have read and agreed to the published version of the manuscript.

**Funding:** Serbian part of this contribution was cofounded through Grant OI 175 014 of the Ministry of Education Science and Technological Development of the Republic of Serbia.

**Data Availability Statement:** Not applicable.

**Acknowledgments:** Not applicable.

**Conflicts of Interest:** The authors declare no conflict of interest.

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
