# Peer review of "Sustainable Health Care Provision Worldwide: Is There a Necessary Trade-Off between Cost and Quality?"

_sustainability, doi:10.3390/su15021372_

Round 1
Reviewer 1 Report
This paper presents an interesting topic in healthcare and sustainability.
The paper can be improved by adding the literature review section and improving the methodology session. The authors should explain why you search only from the google database rather than other database e.g. SCOPUS, ISI.
Then, the research methodology should be thoroughly explained and demonstrated with the flow chart. For now, it still lacks of information on how many papers the authors used to review with this study. Please clearly explain the methodology used for this review.
Please give the rationale on how the subsection 3.1-3.6. were outline.
Author Response
Reviewer 1
Comments
This paper presents an interesting topic in healthcare and sustainability.
Dear Reviewer,
We are grateful with you for your valuable time to examine our paper. Your comments and suggestions significantly contribute to upgrade our paper.
The paper can be improved by adding the literature review section and improving the methodology session. The authors should explain why you search only from the google database rather than other database e.g. SCOPUS, ISI.
Response: We have added relevant papers in methodology sections and wherever it needed. Regarding the literature search, we have different approach. We selected the literature in 4 steps; 1- Search the literatures by key words through Google, 2- Sort the title those best match in first 5 web pages, 3- Go for full access literatures from titles and 4 – Pick up or discard the paper that useful to our title. We had no special preference in database (PubMed, Scholar Google, Scopus, CINHAl, DOAJ etc.) for the full access because most of the paper were multiple database and open access as well. So, display of multiple database flowchart was not meaningful to our title. For example, our one paper “Shadmi, Efrat, Yingyao Chen, Inês Dourado, Inbal Faran-Perach, John Furler, Peter Hangoma, Piya Hanvoravongchai et al. "Health equity and COVID-19: global perspectives." International journal for equity in health 19, no. 1 (2020): 1-16” has indexed many databases; PubMed, PubMed Central, Scopus, DOAJ, Scholar Google etc. On the other hand, one paper is indexed in more than one database, and it creates confusions which database would count. Those 4 steps were sufficient for this title and saved our time, minimize the overlapping issue and problems with multiple indexing. Being a burning public health issue, majority of papers easily available in open access.
Then, the research methodology should be thoroughly explained and demonstrated with the flow chart. For now, it still lacks information on how many papers the authors used to review with this study. Please clearly explain the methodology used for this review.
Response: It is very important, highly expected and useful comments and suggestion to this paper. Firstly, the content and nature of the title is great public health issue and just counting and showing the number of literatures is less important. Likewise, quality health care itself might be multidimensional other than core scientific ground. Usually, the follow chart consists of type of data base, inclusion and exclusion criteria like age, sex, country/community, interventions, specific language, time of publication. Our title does not demand those characteristics we did not show a follow chart. It is mandatory for clinical/community and drug trial, treatment protocols etc. but our topic is burning public health issue and would not limit the above criteria and period of time too. We selected the literatures from peer review papers and open access other than hard selection criteria. The basic features of searching the literature by key words, we have included in summary. We have clearly mentioned in out methodology and limitation of study as well. We realized that putting the follow chart in our topic seems more artificial then necessary, however we have clearly described the core features of flow chart in methodology section.
Please give the rationale on how the subsection 3.1-3.6. were outline
Response: There is no theoretical and other established standard of the outline. We simply presented the outlines from sustainability in health care and fundamental factors for health care equity, access and quality as well. These are basic characteristics by dimensions of health care quality and major determinants of health care quality; quality health care, application of technology and price, health care market, issue of quality health care and price, menu-based health care and quality, sustainable health and health care quality, expectational situations.
The aim of the paper is to identify and explore different dimension of health care quality, examine the association with technology, health care market characteristics. The study has been realized using a narrative review searching by key words based on Google.
Response: We updated the aim of the study in our paper. ‘The aim of this review is to explore the different dimension of health care quality, ex-amine the association with technology, health care market characteristics, additional and optional service of health care, sustainability and some exceptional situations.’
Again, thank you so much valuable insight for this paper and volunteer contribution to scientific research community.
Reviewer 2 Report
The aim of the paper is to identify and explore different dimension of health care quality, examine the association with technology, health care market characteristics. The study has been realized using a narrative review searching by key words based on Google.
Although the work is well structured and explained there are some points that have to be addressed:
Major points:
1) The work of Paré et al (2008), cited by the authors stated that most narrative reviews published in our field do not provide any explanations about how the review process was conducted, and therefore, they are vulnerable on the grounds of subjectivity (Williams, 1998). Although there are no rigid published guidelines that designate how to conduct and report a narrative review, over the years, there have been several efforts to introduce some rigor in their research methodology that will elucidate common pitfalls and bring changes into their publication standards, including a structured approach and transparency in terms of reporting (Cronin et al, 2008).
The authors should clarify their efforts to produce a reliable work: for example, how did they select the output of the Google searches?
The authors should justify more in detail why they chose a narrative review to conduct their research although they know all the drawbacks of such an approach.
2) In the discussion the authors concluded that the patients and consumers believe that high prices are indicators of better quality but there is no difference in quality from supply of provider side. The conclusion seems general, my question is: is this statement still valid, since the two references work are from the 60ths? After covid tsunami maybe the debate is completely reversed. At least the review has to be updated.
Additionally, the work of Barber et al (2021) that discriminates between public or private systems of healthcare which is something that also the authors should include in their work, provides conclusions slightly different from what reported in the paper.
Barber et al (2021) highlight that impact of the different price setting methods on enhancing quality and improving efficiency is small and heterogeneous so more systematic evaluation is needed. So, price setting and health care could be associated.
The authors should discuss this conclusion.
The authors should also correct the citation of the Barber et al (2021) work in their paper.
References:
G. Paré, M-C Trudel, M Jaana and S Kitsiou, Synthesizing information systems knowledge: A typology of literature reviews. Information & Management 2015, 52(2):183-199.
C. Williams, The pitfalls of narrative reviews in clinical medicine, Ann. Oncol. 9 (6), 1998, pp. 601–605
P. Cronin, F. Ryan and M. Coughlan, Undertaking a literature review: a step-by-step approach, Br. J. Nurs. 17 (1), 2008, pp. 38–43.
Y. Levy and T.J. Ellis, A systems approach to conduct an effective literature review in support of information systems research Inf. Sci., 9 (2006), pp. 181-212
S. L. Barber, L. Lorenzoni and Tomas Roubal Price setting for health care services: a taxonomy. Working Paper. WHO Centre for Health Development (2021). https://extranet.who.int/kobe_centre/sites/default/files/WKC_workingpaper_pricesetting_22June.21.pdf
Author Response
Reviewer 2
Comments
Although the work is well structured and explained there are some points that have to be addressed:
General Response
Dear Reviewer,
We really appreciate your great contribution to upgrade our paper and point out the errors in existing paper. We have carefully addressed your concerns and updated accordingly.
Major points:
1) The work of Paré et al (2008), cited by the authors stated that most narrative reviews published in our field do not provide any explanations about how the review process was conducted, and therefore, they are vulnerable on the grounds of subjectivity (Williams, 1998). Although there are no rigid published guidelines that designate how to conduct and report a narrative review, over the years, there have been several efforts to introduce some rigor in their research methodology that will elucidate common pitfalls and bring changes into their publication standards, including a structured approach and transparency in terms of reporting (Cronin et al, 2008).
Response: Thank you for your suggestions and providing insight regarding our paper. We have modified our paper considering your suggestions. We agree that narrative review is little bit subjective than systematic review. The nature of title for our paper is most suited as narrative review because here are no hard inclusion/exclusion criteria (Age, sex, race, country, community, language and time frame) and there might be multiple dimensions of quality health care like patients’ perspective, health care provider, regulatory organizations etc. We accept that there is some space for subjectivity, but we made specific scientific ground of health care quality based on references (Table 1) and key determinants of health care cost in 5 subtopic so that how those factors link with quality health care with surplus price.
The authors should clarify their efforts to produce a reliable work: for example, how did they select the output of the Google searches?
Response: We have presented our work in a reliable way. Only we concisely presented the methodology and escape the follow chart. Moreover, we did not mention artificial nature of database and number of literature we used. We applied the google search engine and did not mentioned the database because there could be multiple database of any paper. We searched the literature by key words, sorted the applicable title and went forward to access the full paper from first 5 pages. Almost all types of research papers, have been accessible by advance search approach. It is related to searching strategies other than reliability of papers. Regarding the quality, almost literatures are peer reviewed journal and all of them available online.
The authors should justify more in detail why they chose a narrative review to conduct their research although they know all the drawbacks of such an approach.
Response: Thank you for your concern. We would say again that our title itself more fit for narrative review other than systematic review because it is not related to specific disease/illness, treatment protocol, clinical trials etc. So, we concluded that for the sustainable health care and general perspective, such narrative review could best for this paper. Sure, we are known about some pitfalls of this review. We have mentioned in our limitation of the study. Remaining things we want to leave to the readers.
In the discussion the authors concluded that the patients and consumers believe that high prices are indicators of better quality but there is no difference in quality from supply of provider side. The conclusion seems general, my question is: is this statement still valid, since the two references work are from the 60ths? After covid tsunami maybe the debate is completely reversed. At least the review has to be updated.
Response: Thank you, we completely agreed. We clearly mentioned that cost/price is mostly influenced by market principles (Normally demands and supply). For example a clients may take as regular, follow up, one time etc. but from provider perspectives, they would equally maintain the standard, flow the same treatment protocols, procedures either who have health insurance, those pay out of pocket or those not able to pay or somethings else. The price/cost is not a matter of health care providers, and it is ethical issue too. The scenario in 60s and now is not profoundly different in the matter of quality and cost of health service and remains in future too. Right now, we have added a reference from Thailand 2020.
Additionally, the work of Barber et al (2021) that discriminates between public or private systems of healthcare which is something that also the authors should include in their work, provides conclusions slightly different from what reported in the paper.
Barber et al (2021) highlight that impact of the different price setting methods on enhancing quality and improving efficiency is small and heterogeneous so more systematic evaluation is needed. So, price setting and health care could be associated. The authors should discuss this conclusion. The authors should also correct the citation of the Barber et al (2021) work in their paper.
Response: Thank you for your comments. We have revised and modified the writing in results, discussion and conclusion.
References:
- Paré, M-C Trudel, M Jaana and S Kitsiou, Synthesizing information systems knowledge: A typology of literature reviews. Information & Management 2015, 52(2):183-199.
- Williams, The pitfalls of narrative reviews in clinical medicine, Ann. Oncol. 9 (6), 1998, pp. 601–605
- Cronin, F. Ryan and M. Coughlan, Undertaking a literature review: a step-by-step approach, Br. J. Nurs. 17 (1), 2008, pp. 38–43.
- Levy and T.J. Ellis, A systems approach to conduct an effective literature review in support of information systems research Inf. Sci., 9 (2006), pp. 181-212
- L. Barber, L. Lorenzoni and Tomas Roubal Price setting for health care services: a taxonomy. Working Paper. WHO Centre for Health Development (2021). https://extranet.who.int/kobe_centre/sites/default/files/WKC_workingpaper_pricesetting_22June.21.pdf
Response: We have used all those reference in appropriate places.
Again, thank you so much for your valuable comments and insight to our paper. We greatly acknowledge your volunteer work
Reviewer 3 Report
The study is very interesting. The theme of the study is relevant and timely. It is a topic that is, in our view, of global concern. Regarding the different parts of the manuscript article, we note that: The introduction is well detailed, the method is also clear and the results and discussion are quite illustrative and well argued. However, from our point of view, one aspect is quite relevant and should necessarily appear, as this aspect is very crucial. This is the relationship between price and quality of care according to social status. This element is fundamental, because the study is about the sustainable provision of health care worldwide.
Another aspect that should necessarily be developed, even if the authors have pointed it out as a limitation of the study, is that the study does not specify the price by given quality standards.
Author Response
Reviewer 3
Comments and Suggestions for Authors
General Response
Dear reviewer,
We provide sincere thanks for your valuable time to examine our paper and we are sure that your comments accelerate the quality of our work.
The study is very interesting. The theme of the study is relevant and timely. It is a topic that is, in our view, of global concern. Regarding the different parts of the manuscript article, we note that: The introduction is well detailed, the method is also clear and the results and discussion are quite illustrative and well argued. However, from our point of view, one aspect is quite relevant and should necessarily appear, as this aspect is very crucial. This is the relationship between price and quality of care according to social status. This element is fundamental, because the study is about the sustainable provision of health care worldwide.
Another aspect that should necessarily be developed, even if the authors have pointed it out as a limitation of the study, is that the study does not specify the price by given quality standards.
Response: Thank you so much for your incredible time to examine this paper. We have updated the paper based on reviewer’s comments and suggestions. As you mentioned we updated the writing as price and quality of care according to social status. Another important, we added the limitation of study as ‘this study does not specify the price by given quality standard.’ Further, we have elaborated our methodology, updated writing, added more useful references, and updated our conclusion. We further described methodology, updated new reference, thoroughly revised the writings.
Reviewer 4 Report
General concern: The manuscript is written in an English that is unacceptable to the reader and needs a thorough revision. Many sentences therefore remain unclear or are misleading.
Abstract: “We found that health care as a service industry, and quality health care is not associated extra price.” This sentence is not complete.
Abstract: “Likewise, there are some additional choices in health care and
those are associated with quality.” I do not understand the contents of that sentence.
Line 38: “There is a great question that quality health care is associated with additional cost? Why using the “?”
Line 40: “…those not able to extra pay would compromise the quality health care [7].” Those who are not able to extra pay would not receive quality health, not compromise health!
Line 43: “In high income countries >10% patients adversely affected during treatment,….” What does this half-sentence mean mean concretely?
Line 47: “…the cost associated with medication errors has been estimated at US$ 42 billion annually, not counting lost wages, foregone productivity or health care costs[11]. Which countries are addressed?
Line 51: “A study by Yaqi Yuan explored from 30 countries that only 17% people satisfied from health care facilities [13]. Where does Mr. Yuan come from? What kind of study has been published?
Line 56: “The organizational system is cyclic,…” What does this mean?
Line 58: “Sylvia 2015 highlighted that continuous upgradation…" Please provide a reference! Sylvia alone – or Sylvia et al.?
Line 63: “A study by Mary Dixon-Woods, Sarah McNicol and Graham Martin explored…” As a rule citations are restricted to the last name of the first author followed by "et al" or "and co-workers"!
Line 203: “The revenue health care industry is projected about 60 billion, annual growth 12%, with a projected market volume of US$18,630.00m in 2022, most revenue is generated in China at the end of 2022 [51].” Worldwide or only in China?
Line 217: “Branning 2016 showed that in spite of the special characteristics of the health care industry, the costs of care would be split as up to 30% in physician cost, 10% pharmaceutical cost, 21% administrative cost and up to
36% in new innovation and technology [54].” Which countries are addressed?
Line 220: “It seems that the administrative cost is average with other industry, cost for innovation and technology obviously pay back and not necessarily increase the price but Ogura 2014 concluded that enhance the quality of care, human resource cost is higher than other industry but due to the competitive situation, can be controlled.” This sentence needs some amendment. What does “average with” mean? Which “industry” is addressed?
Line 237: “There are enough clinical protocols, standards, guidelines in medical practice and world health organization, frequently updates them.”
What´s the meaning of that sentence?
Line 240: “Neff et.al 2009 focused that in those protocols and guidelines, there is no mandatory of health care menu and ultimately it increases
the health disparity [59].”
Why should “health care menu” increase health disparity? Please clarify!
Author Response
Reviewer 4
Comments and Suggestions for Authors
General concern: The manuscript is written in an English that is unacceptable to the reader and needs a thorough revision. Many sentences therefore remain unclear or are misleading.
General Response: We greatly value your comments and suggestion and take very positively. As you provided comments, we have addressed as below for every comment/suggestion.
Response: Thank you for your great time to examine our paper. We have revised the writing, made it simple and readable. Our steps to revise the language: we revised ourselves, improved the writing by native speaker and online check and correction method; QuillBot (https://quillbot.com/grammar-check?
There are different opinions regarding the English language on scientific papers and we respect all opinions. There are different English styles of writing even by countries and communities. According to applications, there are multiple aspects of English language like broadcasting, news/magazine, business/market, literatures, Phonetics, Scientific Writing etc. So, we always focus on making simple and readable without complexity and every reader would understand it considering the basic compositions of writing. Some typos, punctuations, gaps, symbols etc. might not be transfer in different format as submitted by authors and errors would be appear significantly. So, those errors would be perfect until the time of proof reading. Obviously, our aim is to maintain the basic standard of writing. We published many papers, worked as reviewer, editor, and many roles but it is never ending process. We appreciate your comments.
Abstract: “We found that health care as a service industry, and quality health care is not associated extra price.” This sentence is not complete.
Response: Thank you. We splitted this sentences and made it clear.
Abstract: “Likewise, there are some additional choices in health care and
those are associated with quality.” I do not understand the contents of that sentence.
Response: Thank you, we have modified in simple form; there might be some additional choices during certain procedure and those may have different prices and would be link with quality.
Line 38: “There is a great question that quality health care is associated with additional cost? Why using the “?”
Response: We have updated it.
Line 40: “…those not able to extra pay would compromise the quality health care [7].” Those who are not able to extra pay would not receive quality health, not compromise health!
Response: We updated it as your suggested version.
Line 43: “In high income countries >10% patients adversely affected during treatment,….” What does this half-sentence mean mean concretely?
Response: Thank you, there were some problems during in transferring the sentences in another version. We corrected those incomplete sentences.
Line 47: “…the cost associated with medication errors has been estimated at US$ 42 billion annually, not counting lost wages, foregone productivity or health care costs[11]. Which countries are addressed?
Response: It was taken from National Strategic Plan on antimicrobial resistant 2017-21, Thailand. We have updated data and resource.
Line 51: “A study by Yaqi Yuan explored from 30 countries that only 17% people satisfied from health care facilities [13]. Where does Mr. Yuan come from? What kind of study has been published?
Response: It was the finding from International Social Survey Program 2011, 32 countries and 34212 respondents. We have clarified in paper.
Line 56: “The organizational system is cyclic,…” What does this mean?
Response: Organization set up yearly program and at the end of every year as cycle, there is a need continuous improvement and update/revised. It is in a cycle system. We have clarified it.
Line 58: “Sylvia 2015 highlighted that continuous upgradation…" Please provide a reference! Sylvia alone – or Sylvia et al.?
Response: We have updated that line Sylvia et al. 2015.
Line 63: “A study by Mary Dixon-Woods, Sarah McNicol and Graham Martin explored…” As a rule citations are restricted to the last name of the first author followed by "et al" or "and co-workers"!
Response: We have revised accordingly.
Line 203: “The revenue health care industry is projected about 60 billion, annual growth 12%, with a projected market volume of US$18,630.00m in 2022, most revenue is generated in China at the end of 2022 [51].” Worldwide or only in China?
Response: We have revised as the revenue health care industry worldwide is projected about 50 billion, annual growth 12%, with a projected market volume of US$15,830.00m, and most of the revenue is generated in China at the end of 2022 [51].
Line 217: “Branning 2016 showed that in spite of the special characteristics of the health care industry, the costs of care would be split as up to 30% in physician cost, 10% pharmaceutical cost, 21% administrative cost and up to 36% in new innovation and technology [54].” Which countries are addressed?
Response: It is in the USA. We have mentioned in revised paper.
Line 220: “It seems that the administrative cost is average with other industry, cost for innovation and technology obviously pay back and not necessarily increase the price but Ogura 2014 concluded that enhance the quality of care, human resource cost is higher than other industry but due to the competitive situation, can be controlled.” This sentence needs some amendment. What does “average with” mean? Which “industry” is addressed?
Response: We have modified the writing as ‘It seems that the administrative cost of health care is similar with other industries, cost for new innovation and technology obviously pay back and not necessarily increase the price but a review by Ogura et.al 2014 concluded that to enhance the quality of care, human resource cost is higher than other industries but somehow, it can be balanced by appropriate supply chain mechanism [55]. So, it is not the main issue of price increase, but it is the issue of cost adjustment in health care industry’. It is simple and clear to understand.
Line 237: “There are enough clinical protocols, standards, guidelines in medical practice and world health organization, frequently updates them.” What´s the meaning of that sentence?
Response: We have revised as ‘there are specific clinical procedures, protocols, standards, guidelines in medical practice and World Health Organization (WHO), frequently updates them. Heymann 1994 highlighted that those clinical protocols are keys to maintain quality health care and adopting those protocols is the professional ethic of health care providers.’
Line 240: “Neff et.al 2009 focused that in those protocols and guidelines, there is no mandatory of health care menu and ultimately it increases the health disparity [59].” Why should “health care menu” increase health disparity? Please clarify!
Response: We have modified the sentences and provided the reference that choice-based health care creates economic class, encourages selection bias and catastrophic health expenditure.
We again, provide our respect for your scrutiny toward our paper.
Round 2
Reviewer 2 Report
The paper now is fine.
Reviewer 4 Report
Well done!